# Diagnostic and Therapeutic Characteristics in Patients with Pneumotorax Associated with COVID-19 versus Non-COVID-19 Pneumotorax

**DOI:** 10.3390/medicina58091242

**Published:** 2022-09-08

**Authors:** Claudiu-Eduard Nistor, Daniel Pantile, Camelia Stanciu-Gavan, Adrian Ciuche, Horatiu Moldovan

**Affiliations:** 1Department of Thoracic Surgery, Central Military Emergency University Hospital, 013058 Bucharest, Romania; 2Faculty of General Medicine, “Carol Davila” University of Medicine and Pharmacy, 010825 Bucharest, Romania

**Keywords:** COVID-19, pneumothorax

## Abstract

*Introduction*: Pneumothorax is a condition that usually occurs in thin, young people, especially in smokers. It is an unusual complication of COVID-19 disease that can be associated with worse results. This disease can occur without pre-existing lung disease or without mechanical ventilation. *Materials and Methods*: We present a monocentric comparative retrospective study of diagnostic and treatment analysis of two groups of patients diagnosed with COVID-19 and non-COVID-19 pneumothorax. All patients included in this study underwent surgery in a thoracic surgery department. The study was conducted over a period of 18 months. It included 34 patients with COVID-19 pneumothorax and 42 patients with non-COVID-19 pneumothorax. *Results*: The clinical symptoms were more intense in patients with COVID-19 pneumothorax. We found that the patients with COVID-19 had significantly more respiratory comorbidities. Diagnostic procedures include chest CT exam for both groups. Laboratory findings showed that increasing values for the analyzed data were consistent with the deterioration of the general condition and the appearance of pneumothorax in the COVID-19 group. The therapeutic attitude regarding the non-COVID-19 group was to eliminate the air from the pleural cavity and surgical approach to the lesion that determined the occurrence of pneumothorax. The group of patients with COVID-19 pneumothorax received systemic treatment, and only minimal pleurotomy was performed. The surgical approach did not alter patients’ survival. *Conclusions*: Careful monitoring of the patient’s clinic and laboratory tests evaluating the degradation of the lung parenchyma, correlated with the imaging examination (chest CT) is mandatory and reduces COVID-19 complications. Early imaging examination starts an effective diagnosis and treatment management. In severe COVID-19 pneumothorax cases, the pneumothorax did not influence the evolution of COVID-19 disease. When we found that the general condition worsened with the rapid progression of dyspnea and the deterioration of the general condition, and we found that it represented the progression or recurrence of pneumothorax.

## 1. Introduction

Pneumothorax is defined as a collection of air in the pleural cavity [1]. This may become a major surgical emergency. Becoming an emergency depends on factors like the amount of air accumulated in the pleural cavity, which can cause disorders of respiratory and cardiovascular function [2]. Since the onset of SARS-CoV-2 infection and COVID-19 disease, it has been found that in 1–2% of patients with COVID-19 pneumonia, pneumothorax requiring hospitalization can be encountered [3]. Complications such as severe pneumonia, pulmonary edema [4], and pneumothorax [5] are described during COVID-19 disease. Pneumothorax is known to complicate diffuse lung infections [6] and is a common complication of mechanical ventilation during a critical illness [7]. In published studies, pneumothorax has been reported as a complication of COVID-19 [8], with an incidence of 1% in hospitalized patients [4], 3% in hospitalized patients with pneumonia [9], 6% in mechanically ventilated patients [10], and 1% in deceased patients [11]. The etiology is not well known, but known predisposing factors are smoking, intense exertion, cough, sneezing, and defecation effort, which trigger the rupture of emphysema bubbles. Some experts believe that the main risk factor for pneumothorax is smoking tobacco cigarettes [12]. In the case of SARS-CoV-2 infection, if patients also have an associated lung pathology, there is a predisposition for pneumothorax, especially in those intubated and ventilated for a longer period [3].

We have proposed in this paper to perform a monocentric comparative retrospective analysis of patients with non-COVID-19 pneumothorax and COVID-19 pneumothorax in terms of diagnosis and therapeutic attitude.

## 2. Materials and Methods

A retrospective analysis of patients with pneumothorax in COVID-19 and non-COVID-19 disease was performed in terms of diagnostic and therapeutic characteristics using the hospital database. The study was approved by the research ethics commission no. 441/3.03.2021.

All patients who had pneumothorax associated or not with COVID-19, hospitalized in our hospital between July 2020 and 31 December 2021 were included in this study.

Exclusion criteria: patients who had cervicothoracic air accumulations—pneumomediastinum and pneumothorax and/or mixed thoracic effusions (air and liquid) associated or not with the COVID-19 disease.

A total of 34 patients with COVID-19-associated pneumothorax (from 1543 patients admitted with COVID-19) and 42 patients with non-COVID-19 pneumothorax (from 1362 patients admitted in our service) were selected according to the inclusion and exclusion criteria.

Laboratory findings such as white blood cells (WBC) count, D_Dimer, Ferritin, CRP (C-reactive protein), LD-P (lactate dehydrogenase), and γGT (gamma-glutamyl transferase) were compared, as they have a specific variation for patients without COVID-19 and for patients with COVID-19.

Comparative imaging evaluation was performed by analyzing computed tomography examinations in both groups of patients. All patients underwent a CT scan.

Clinical symptoms were also analyzed, and the therapeutic attitude was established according to the patient’s condition in each study group. All patients underwent surgery.

The therapeutic attitude for both groups was to eliminate the air from the pleural cavity and surgically approach the lesion that determined the occurrence of pneumothorax (if possible). The group of patients with COVID-19 pneumothorax also received systemic COVID-19 therapy (kaletalopinavir/ritonavir, azithromycin, low molecular weight heparin, and intravenous immunoglobulin).

For statistical analysis, the data was gathered and organized using Microsoft Excel^®^ (Microsoft Office 365 Suite^®^), and statistical analysis (descriptive and comparative) was performed using the same suite, with several add-ins (R. Fitch Software^®^ WinStat version 14.0, AddinSoft^®^ XLStat version 2022). The results were displayed using tables and graphs created using the same software. Statistical tests applied include a comparison of means (when mean values were statistically analyzed), an odds ratio for the statistical analyzing how much higher the odds of exposure among case-patients are than among controls, Student’s *t*-test to compare two sets of quantitative data.

## 3. Results

We schematically present the results obtained comparatively on each studied group in the following Tables. Thus, we found that age was significantly higher in patients with COVID-19 pneumothorax (30–84 y-o, with an average of 58.29 y-o), when compared with patients without COVID-19 (15–68 y-o, with an average of 41.13 y-o), the mean age being higher, statistically significant, in patients with pneumothorax and COVID-19 (*p* < 0.0001) (Table 1). The most affected patients were males, with a M:F ratio of 26:8 in the study group and 33:9 in the control group (Table 1). In patients with COVID-19, it was found that the presence of pneumothorax is more common in patients over 50 years old (*p* < 0.001).

The incidence of pneumothorax was 2.2% (34 patients out of 1543 patients admitted with COVID-19) in the COVID-19 group, while for the non-COVID-19 group, the incidence was 3% (42 out of 1362 patients).

All patients had a sudden onset of dyspnea, accompanied by respiratory distress and sometimes chest pain; fever, and persistent cough were more common in COVID-19 patients.

The evolution of patients was favorable for all non-COVID-19 patients, while the vast majority of positive COVID-19 patients had an unfavorable outcome.

Patients with non-COVID-19 pneumothorax had significantly fewer comorbidities, but all were smokers (Table 2). While all patients with pneumothorax and COVID-19 had at least one comorbidity, in the control group, only 62% (*n* = 26) of the patients had at least one comorbidity (*p* = 0.009). When analyzing the smoking status, in the studied group 47% (*n* = 16) were smokers, compared to 80% (*n* = 34) in the study group (*p* = 0.002).

We also analyzed the presence of other respiratory comorbidities in the two groups (COPD – chronic obstructive pulmonary disease, asthma, bronchiectasis, pulmonary emphysema) and found that the patients with COVID-19 had significantly more respiratory comorbidities (Table 2). The same results were found when we analyzed the cardio-vascular comorbidities (Table 2). 

Regarding the laboratory findings, in non-COVID-19 patients, the values were almost normal, while for patients with COVID-19, most of them are outside the normal range (Table 3). All these differences were statistically significant (Table 3). Analyzing these results, we can conclude that all the abnormal laboratory findings were due to the COVID-19 disease and not a consequence of pneumothorax. Increasing values for these laboratory tests were consistent with the deterioration of the general condition and the appearance of pneumothorax in the COVID-19 group.

Chest X-Ray was performed in all patients with non-COVID-19 pneumothorax, and CT examination was performed for all patients in both study groups (Figure 1). A very small number of patients (4 patients) benefited from ultrasound examination, all from the group of pneumothoraxes with COVID-19. With no statistical significance, we found that pneumothorax was more often on the right side in both studied groups. The incidence of right pneumothorax was 73.5% in patients with COVID-19 and 54.7% in patients without COVID-19 (Table 3).

The symptoms were more intense in patients with COVID-19 pneumothorax. This group included: 21 patients with pneumothorax after intubation and mechanical ventilation, 9 patients with pneumothorax while on CPAP ventilatory support, and 4 patients with pneumothorax without ventilatory support. The group of patients with non-COVID-19 pneumothorax had sudden onset of dyspnea and chest pain.

Fever and chest pain were significantly more frequent in patients with COVID-19 (*p* < 0.0001, respectively 0.01) (Table 4).

In the group of non-COVID-19 pneumothorax, we performed video assisted thoracic surgery techniques (VATS) associated with chemical pleurodesis with iodine, parietal apical pleurectomy and mechanical resection of emphysema bubbles. The surgery for COVID-19 group consisted of performing a minimal pleurotomy with the insertion of a chest tube to evacuate the air. The system was connected to a Béclère-type drainage. VATSsurgery was considered not suitable for patients with underlying COVID-19 disease; thus, none of them was treated using VATS procedures (Table 5).

All patients with non-COVID pneumothorax were cured after surgery, while in the group of COVID-19 pneumothorax, 24 patients died within 30 days of surgery, and 10 patients were discharged (4 patients were discharged with the chest tube in place, 4 patients were discharged completely cured, and we do not have postoperative data for 2 patients).

## 4. Discussion

Considering the relatively small number of patients included in this study, over the described period of time, the two study groups could not be matched for the number of patients included in each group, age, and comorbidities. Furthermore, for the first patients included in the study, testing the sub-variant of SARS-CoV-2 virus was not available; thus, we decided the COVID-19 group would span over multiple waves and variants of the virus. When testing the sub-variant of the virus became available, this test was included, but due to the small number of patients, we decided to use this data in a follow-up of this study.

Imaging studies confirm the diagnosis of pneumothorax—chest radiography, computed tomography, and chest ultrasound [13]. Radiological examination is mandatory to highlight the air in the pleural cavity, with the absence of lung pattern [14]. Chest computed tomography highlights the triggering cause [lung bullae, lung emphysema, tumors, etc.), specifying the topographic location when dealing with a partial pneumothorax; it can also differentiate between an air cyst and a giant emphysema blister, estimating the effective size of a pneumothorax and the diagnosis of other underlying lung diseases [12,15]. In our study, all patients with non-COVID pneumothorax benefited from radiological examination; the CT examination was performed for both study groups (with non-COVID-19 pneumothorax and those with COVID-19). During the last decade, pulmonary ultrasonography has emerged as a sensitive technique in the evaluation of respiratory diseases and has gained a well-established role in the diagnosis of pneumothorax [3,16,17]. Our study group included 4 patients who benefited from the ultrasonographic examination.

Regarding the evaluation of laboratory findings, the literature presented the follow-up of LD-P, D_Dimer, Ferritin, CRP, LD-P, γGT [18], and in some patients the presence of leukopenia [19,20]. Our findings support the values described in the literature. Laboratory results were within normal limits for patients with non-COVID-19 pneumothorax, while for patients with COVID-19 pneumothorax abnormal findings were regarding the values of leukocytes, D_Dimer, Ferritin, CRP, LD-P, γGT.

### 4.1. Treatment Methods for the Pleural Space Management

**Conservative management** is suitable and has indications for patients with minimal symptoms [21]. Indication for this type of treatment is given to patients with a small pneumothorax (<15%) that are clinically healthy and can be kept under observation. Oxygen therapy can be administered at a flow rate of up to 10 L/min (with caution in patients with COPD) [22] and analgesia [23]. One disadvantage is the frequent recurrences and prolonged healing time. In patients with pneumothorax and COVID-19 we followed very closely the increase in the volume of intrapleural air, and no patient could be treated conservatively.

### 4.2. Interventional Management (Medical–Surgical Therapeutic Attitude)

**Management for outpatient treatment**—**air aspiration**—(exsufflation) is considered to be the first therapeutic gesture in pneumothoraxes with indications for air evacuation. Air aspiration is performed with a needle, followed by a chest X-Ray some 2–4 h after the maneuver [15]. Success rates are estimated at 51–69% [24]. This therapeutic approach is suited only for patients with a first episode of pneumothorax greater than 15%, and up to 30%, who are hemodynamically stable [14,23]. The advantages of this procedure are represented by the simple technique and the low cost. The possible infection of the pleural cavity and the risk of damaging the lung parenchyma during the maneuver are considered its most important disadvantages [15]. This maneuver could not be applied to the patients included in this study, however, its effectiveness is proven only on spontaneous pneumothoraxes [14,23].

**Management of pleural decompression by mounting chest tube**—minimal pleurotomy. This technique of minimal pleurotomy consists in placing a chest drainage tube in the pleural cavity through an intercostal space [25], usually on the 5th intercostal space, on the middle axillary line (Bulau-type minimal pleurotomy) [14]. The drain tube is maintained for 3–5 days. It is attached to a one-way Heimlich [26]/Vygon valve [27] or to a suction device. Suction drainage can be passive (Béclère kit) or active (suction battery) [1,25]. The chest tube can be single or double lumen (a double-lumen chest tube is used for performing chemical pleurodesis through the tube) [28]. Minimal pleurotomy is performed in patients with altered general conditions and in those who do not undergo general anesthesia for VATS. It is usually indicated in a maximum emergency when the patient has severe cardiorespiratory insufficiency—massive hypertensive pneumothorax, with the presence of the “chake-valve” mechanism, or in the case of bilateral pneumothorax [14]. It is not indicated for patients with pneumothorax caused by rupture of emphysema bullae. The main disadvantage of this method is the relatively high recurrence rate of the disease 23–50% [29].

In the group of patients with COVID-19 pneumothorax, only minimal pleurotomy was performed without chemical pleurodesis. In 12 patients in the non-COVID-19 pneumothorax group, the aerial effusion was remitted after minimal pleurotomy with chemical pleurodesis on the tube. No further surgery was required.

**Chemical pleurodesis** aims to achieve a pleural symphysis to prevent recurrences [14]. International guidelines suggest chemical pleurodesis, both through the chest tube and by minimally invasive surgical techniques, to achieve pleuro-pulmonary adhesion [28] and to prevent recurrence [30]. The chemical agents known for pleurodesis are talcum powder [31] and iodine [28], autologous blood, silver nitrate, tetracycline, and doxycycline [30]. The main disadvantage of chemical pleurodesis is the difficulty of performing another surgery on the same side in case of subsequent lung disease [28,32]. The administration of the chemical agent through the chest tube also has another disadvantage: only local action on the trajectory of the tube holes and not on the entire pleural surface.

In the studied groups, only chemical pleurodesis with betadine was used in non-COVID-19 pneumothorax patients. We did not use talcum powder because this type of chemical pleurodesis has several side effects.

**Video Assisted Thoracoscopic Surgery (VATS)**—is considered the main approach for the treatment of pneumothorax and for performing pleurodesis (a pleural inflammatory process with fibrous adhesions to prevent recurrence) [23]. Using this approach, the surgeon is able to inspect the entire pleural cavity, detecting [14] and resolve the pulmonary lesions, and also perform chemical pleurodesis with fibrotic effect, with suppression of the pleural space [31] and efficient drainage of the pleural cavity [33]. According to systematic reviews of controlled studies, the effects of pleural abrasion and chemical pleurodesis have been shown to be effective through VATS [34].

Chemical pleurodesis with iodine using a VATS approach is superior to pleurodesis through the chest tube because the sclerosing agent can be evenly distributed, under video control, over the entire surface of the lung parenchyma [28].

In patients with SARS-CoV-2 infection, minimal pleurotomy was the method of choice for resolving pneumothorax, and it must be performed with great care, as there is a danger of generating aerosols and infecting the physician performing this maneuver [3]. If the pneumothorax is due to an emphysema blister, therapeutic management should be directed to minimally invasive surgery, with a surgical resolution of the emphysema blister (ligation at its base or its resection) associated with mechanical and/or chemical pleurodesis. Therefore, it is important that the medical personnel have adequate protective equipment during the insertion of the chest tube, its connection to the drainage circuit and the use of filters to limit the spread of the virus [35,36]. Understanding the mechanism of association between COVID-19 and pneumothorax is necessary to apply the correct methods to prevent the occurrence of pneumothorax.

Other surgical procedures [14] usually used are excision/ligation at the base of the lung bullae, associated or not with apical pleurectomy; wedge resections of the affected lung area—resection of blisters.

In the studied groups, we performed therapeutic management oriented towards managing acute respiratory failure and acute respiratory distress. Emergency drainage of the pleural space was the optimal therapeutic option in the pneumothorax COVID-19 group. In the non-COVID-19 group, we performed VATS with apical pleurectomy, chemical pleurodesis, and resection of lung bullae, and we did not register any recurrences.

Considering the relatively small number of patients, as we consider this to be one of the study’s limitations, the two study groups are not matched for the number of patients included in each group, age, and comorbidities. The COVID-19 group also spans multiple waves and variants of the virus, as the detection of the virus did not include the virus’ sub-variant analysis.

## 5. Conclusions

The pathophysiological mechanisms for pneumothorax, although not well established, may be due to SARS-CoV-2 infection. Regardless of the association of pneumothorax with COVID or the non-association of these diseases, pneumothorax remains a surgical emergency.

We found that careful monitoring of the patient’s clinical status of laboratory tests evaluating the degradation of the lung parenchyma, correlated with the imaging examination, is necessary to establish the evolution of these patients. We used ultrasound examination in four COVID-19 patients because it could be performed at the patient’s bed, and it was no longer necessary moving the patient to the CT examination laboratory.

Careful monitoring of these patients may also reduce COVID-19 complications and mortality (being a small sample, we cannot fully conclude that careful monitoring reduces morbidity and mortality for these patients). Early imaging examination starts an effective diagnosis and treatment management. In severe COVID-19 pneumothorax cases, the pneumothorax did not influence the evolution of COVID-19 disease. In the group of non-COVID-19 patients, the postoperative evolution was favorable. Postoperative evaluation was performed only by imaging examination.

In the COVID-19 pneumothorax group, when we found that the general condition worsened with the rapid progression of dyspnea and the deterioration of the general condition, we also found that it represented the progression or recurrence of pneumothorax.

In patients with COVID-19 and pneumothorax, minimal pleurotomy was the method of choice for resolving it, as there was no other underlying condition posing a risk for pneumothorax. For the group without COVID-19, pneumothorax was a consequence of emphysema blister rupture, and therapeutic management should be directed towards minimally invasive surgery, with a surgical resolution of the emphysema blister (ligation at its base or its resection) associated with mechanical and/or chemical pleurodesis.

## Figures and Tables

**Figure 1 medicina-58-01242-f001:**
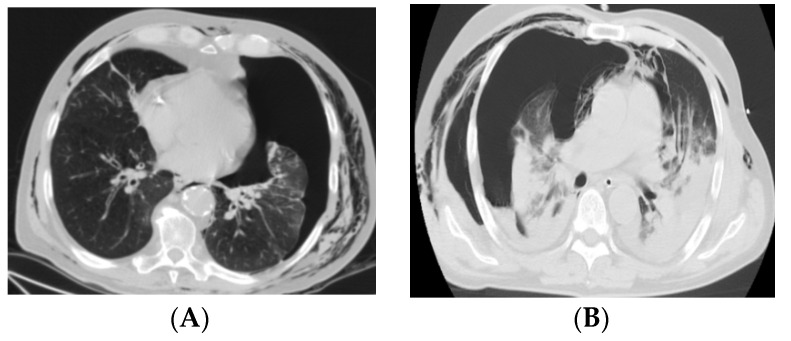
**CT images for COVID-19 and non-COVID-19 pneumothorax:** (**A**) left pneumothorax in a patient without COVID-19; (**B**) right pneumothorax in a patient with COVID-19.

**Table 1 medicina-58-01242-t001:** Characteristics of the study groups.

	COVID-19 Associated Pneumothorax *n* = 34	Pneumothorax without COVID-19 *n* = 42	*p*-Value
Age variation (y-o)	30–84	15–68	N/A
Mean age (y-o)	58.29	41.13	<0.0001 *
Sex ratio (M:F)	26:8	33:9	N/A

* statistical test applied: comparison of means.

**Table 2 medicina-58-01242-t002:** Analysis of comorbidities per patient group.

Comorbidities	COVID-19 Associated Pneumothorax *n* = 34	Pneumothorax without COVID-19 *n* = 42	Statistic Test	*p*-Value
No comorbidity	0 (0%)	16 (38.09%)	OR = 0.023	0.009
Smoker	16 (47.05%)	34 (80.95%)	OR = 0.209	0.002
COPD	21 (61.76%)	0 (0%)	OR = 135.37	0.0008
Asthma	6 (17.64%)	0 (0%)	OR = 19.38	0.04
Bronchiectasis	26 (76.47%)	0 (0%)	OR = 265	0.0002
Systemic hypertension	18 (52.94%)	0 (0%)	OR = 95.3	0.001
Pulmonary emphysema	5 (14.7%)	26 (61.9%)	OR = 0.106	0.0001
Tachycardia	14 (41.17%)	0 (0%)	OR = 60.12	0.005
Heart disease	16 (47.05%)	0 (0%)	OR = 75.81	0.003

OR–odds ratio.

**Table 3 medicina-58-01242-t003:** Comparative values of laboratory findings and imaging diagnosis.

	COVID-19 Associated Pneumothorax *n* = 34		Pneumothorax without COVID-19 *n* = 42		Statistic Test	*p*-Value
**Haematology**	**Range**	**Average**	**Range**	**Average**		
WBC (k/microL)	4.20–24.32	14.3	3.80–8.76	7.8	*t*-test = 10.29	<0.0001
D_Dimer (ng/mL)	305–720	505	0–250	125	*t*-test = 16.4	<0.0001
**Blood chemistry**	**Range**	**Average**	**Range**	**Average**		
Ferritin (mg/dL)	356–833	606	20.00–250	140	*t*-test = 20.11	<0.0001
CRP (mg/L)	28.08–71.3	58.1	0.00–5.00	3	*t*-test = 15.44	<0.0001
LD-P (U/L)	283.69–387.07	303.8	125.00–220.00	176.2	*t*-test = 4.55	<0.0001
γGT (U/L)	63.02–98.75	74.6	11.00–59.00	23.9	*t*-test = 4.94	<0.0001
**Imaging Diagnosis**						
Right pneumothorax	25		23		OR = 2.29	0.09
Left pneumothorax	9		19		OR = 0.43	0.09

**Table 4 medicina-58-01242-t004:** Patients’ symptoms.

	COVID-19 Associated Pneumothorax *n* = 34	Pneumothorax without COVID-19 *n* = 42	Statistic Test	*p*-Value
Dyspnea	34 (100%)	42 (100%)	OR = 0.81	0.91
Persistent cough	34 (100%)	1 (2.38%)	OR = 2.49	0.57
Fever	34 (100%)	0 (0%)	OR = 5865	<0.0001
Chest pain	23 (67.64%)	42 (100%)	OR = 0.024	0.01

**Table 5 medicina-58-01242-t005:** Therapeutic attitude.

Therapeutic Attitude	COVID-19 Associated Pneumothorax *n* = 34	Pneumothorax without COVID-19 *n* = 42	Statistic Test	*p*-Value
Chest tube drainage	34	12	OR = 168.36	0.0005
Video assisted (VATS) procedures *	0	30	OR = 0.005	0.0005

* wedge resections, chemical pleurodesis, apical pleurectomy, resection of emphysema bullae, etc.

## Data Availability

The data presented in this study are available on request from the corresponding author. The data are not publicly available due to the privacy policy of the centers involved in the study.

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
