# Peer review of "Diagnostic and Therapeutic Characteristics in Patients with Pneumotorax Associated with COVID-19 versus Non-COVID-19 Pneumotorax"

_medicina, 2022, doi:10.3390/medicina58091242_

Round 1

Author Response

Thak you for your review. We have edited the text in accordance with most of your requirements.

Reviewer 2 Report

The authors have done a good job of describing diagnostic and treatment analysis of patients diagnosed with COVID-19 and non-15 COVID-19 pneumothorax. The manuscript is well written. The study design is appropriate and the conclusion drawn from the results seems correct.

Author Response

Thak you for your review. 

Reviewer 3 Report

Insufficient description in Materials and Methods – information about statistical tests, used software are missed, poor information about laboratory analyses.

A Results are below the scientific level. They look like laboratory report.

A Discussion presents mainly descriptions of pneumothorax treatment and interventional methods and is far from the Results.

English language correction is needed (e.g. “significantly higher” instead of “significantly bigger”, grammar mistakes).

Many abbreviations are not explained in the text.

Author Response

(The authors gave the same response as above.)

Round 2

Reviewer 1 Report

It's good to see that the issues highlighted are addressed accordingly making the manuscript good to read. Please correct the references as per Vancouver style; hyphens in between author names to be replaced by commas with full stop after the last author name.

Rest I am happy to accept for publication

Author Response

Dear reviewer,

We have addressed the references and author names, complying with your indications. Thank you.